# Gene Expression Signatures of a Preclinical Mouse Model during Colorectal Cancer Progression under Low-Dose Metronomic Chemotherapy

**DOI:** 10.3390/cancers13010049

**Published:** 2020-12-26

**Authors:** Hung Ho-Xuan, Gerhard Lehmann, Petar Glazar, Foivos Gypas, Norbert Eichner, Kevin Heizler, Hans J. Schlitt, Mihaela Zavolan, Nikolaus Rajewsky, Gunter Meister, Christina Hackl

**Affiliations:** 1Biochemistry Center Regensburg (BCR), Laboratory for RNA Biology, University of Regensburg, 93053 Regensburg, Germany; Xuan-Hung.Ho@vkl.uni-regensburg.de (H.H.-X.); Gerhard.Lehmann@vkl.uni-regensburg.de (G.L.); Norbert.Eichner@vkl.uni-regensburg.de (N.E.); Kevin.Heizler@vkl.uni-regensburg.de (K.H.); 2Laboratory for Systems Biology of Gene Regulatory Elements, Berlin Institute for Medical Systems Biology, Max-Delbruck Center for Molecular Medicine, 10115 Berlin, Germany; petar.glazar@mdc-berlin.de (P.G.); rajewsky@mdc-berlin.de (N.R.); 3Biozentrum, University of Basel, 4056 Basel, Switzerland; Foivos.Gypas@fmi.ch (F.G.); mihaela.zavolan@unibas.ch (M.Z.); 4Department of Surgery, University Hospital Regensburg, 93053 Regensburg, Germany; Hans.Schlitt@ukr.de

**Keywords:** LDM topotecan chemotherapy, miRNA, interferon-induced genes, lncRNAs, LINC00483, MIR210HG, micropeptide, circRNA, circZNF609, colorectal cancer

## Abstract

**Simple Summary:**

Colorectal cancer is one of the most frequent types of cancer world-wide, leading to over 500,000 cancer-related deaths each year. Although many primary colorectal cancer patients can be cured by surgery, up to 60% will develop metastases. Chemotherapeutic strategies are well-established, but finally often lead to chemo-resistance and tumor relapse. A specific chemotherapeutic approach is low dose metronomic (LDM) therapy, which is based on a constant administration of low doses of a chemotherapeutic compound instead of high-dose pulses, which are often a huge burden for patients and also may induce rapid resistance. However, the molecular mechanism of LDM chemotherapy is not fully understood. Our study therefore aims at identifying gene signatures of colorectal cancer progression under LDM chemotherapy which eventually provides new potential biomarkers for therapeutic interventions.

**Abstract:**

Understanding the molecular signatures of colorectal cancer progression under chemotherapeutic treatment will be crucial for the success of future therapy improvements. Here, we used a xenograft-based mouse model to investigate, how whole transcriptome signatures change during metastatic colorectal cancer progression and how such signatures are affected by LDM chemotherapy using RNA sequencing. We characterized mRNAs as well as non-coding RNAs such as microRNAs, long non-coding RNAs and circular RNAs in colorectal-cancer bearing mice with or without LDM chemotherapy. Furthermore, we found that circZNF609 functions as oncogene, since over-expression studies lead to an increased tumor growth while specific knock down results in smaller tumors. Our data represent novel insights into the relevance of non-coding and circRNAs in colorectal cancer and provide a comprehensive resource of gene expression changes in primary tumors and metastases. In addition, we present candidate genes that could be important modulators for successful LDM chemotherapy.

## 1. Introduction

Low-dose metronomic (LDM) chemotherapy, a regular administration of chemotherapeutic agents at low doses with close intervals over prolonged periods of time, is an emerging form of cancer therapy [1,2,3]. In the last 20 years, several regimes of metronomic therapy have been applied to multiple cancer types, especially breast cancer [4,5,6] and many ongoing LDM clinical trials at different phases are currently registered to the database of U.S. National Institutes of Health (www.clinicaltrials.gov) [7]. However, similar to other anticancer therapies, LDM chemotherapy finally often results in the development of resistance [8]. Numerous preclinical studies have identified molecular targets and pathways to understand the mechanism of LDM chemotherapy [1]. So far, known mechanisms of LDM chemotherapy include suppression of tumor-angiogenesis [1,9,10], depletion of regulatory T cells [10], direct tumor cell targeting effects [11,12] and the induction of tumor dormancy [2]. Some mechanisms that drive LDM chemotherapy resistance have been described. The detailed mechanism, however, is yet unknown [8,13,14]. 

LDM chemotherapy has been applied to colorectal cancer [15,16,17,18]. Previously, we could show that LDM topotecan significantly reduces liver metastasis and prolongs survival in a preclinical orthotopic model of colon cancer [19]. Topotecan is an inhibitor of the DNA-topoisomerase 1. The same mechanism of action is used by irinotecan, which is one of the standard chemotherapeutic drugs given to colorectal cancer patients. The reason why we used topotecan instead of irinotecan is, that topotecan can be given orally, which is a great benefit for a daily, low-dose application both in preclinical trials as well as in patient treatment. Other studies using microarray-based analyses have identified several genes that are associated with the response to and resistance of topotecan in ovarian cancer [20,21]. However, little is known about the molecular basis of LDM chemotherapy in colorectal cancer and especially the roles of regulatory RNAs such as microRNAs (miRNAs), long non-coding RNAs (lncRNAs) or circular RNAs (circRNAs) remain elusive. 

MiRNAs represent a large and well-studied class of small non-coding RNAs that bind to the 3′ untranslated region of target mRNAs and inhibit their expression [22,23]. MiRNA expression is dysregulated in several diseases, including cancer [24,25,26]. Depending on their target repertoire, miRNAs can function as tumor suppressors or oncogenes and have been suggested as promising prognostic indicators [25,26]. Interestingly, miRNAs stably circulate in the blood stream and other body fluids, further strengthening their potential use as biomarkers in liquid biopsies [27]. miRNAs not only regulate specific target mRNAs, but also interact with other RNA species including lncRNAs, circRNAs or pseudogene transcripts, to form complex regulatory networks [28]. Detailed knowledge of such networks is a prerequisite for understanding complex cellular programs underlying cancer manifestation, metastatic progression and therapeutic success [29]. 

Comprehensive genome-wide gene expression studies have identified thousands of lncRNAs with largely unknown functions. Many of these lncRNAs seem to be mis-expressed in cancers [30,31]. LncRNAs are also dysregulated during chemotherapy treatment and drug resistance [32,33,34] and may serve as potential therapeutic targets [35,36]. For example, the urothelial cancer-associated 1 (*UCA1*) lncRNA is upregulated in cisplatin-resistant bladder cancer, which activates Wnt signaling in a Wnt6-dependent manner [34]. Furthermore*, NEAT1* is a p53-induced lncRNA [37] and activation of p53 stimulates the formation of *NEAT1* paraspeckle, which are nuclear structures related to gene expression activity [38,39]. Moreover, a specific *NEAT1* isoform (*NEAT1_2*) can predict the response of platinum-based chemotherapy [38]. An example of a complex regulatory network is represented by the double-negative feedback loop between the lncRNA *MIR100HG*, the *GATA6* mRNA and miR-125b in the regulation of Wnt signaling, which is associated with cetuximab resistance in colorectal cancer [33]. miR-100 and miR-125b mediate cetuximab resistance via repression of several Wnt inhibitors, which consequently leads to an increased Wnt signaling. Interestingly, miR-125b negatively regulates the expression of the transcription factor GATA6. In turn, GATA6 binds to GATA-binding sites in the promoter region of *MIR100HG* and represses the expression of *MIR100HG*. These regulatory networks provide a molecular basis for understanding the complexity of cetuximab resistance and highlight the importance of such networks for successful cancer therapy [33].

A number of studies have found that some of the annotated lncRNAs encode, in fact, small peptides [40,41]. This is because the algorithms for finding and annotating open reading frames (ORF) typically use a cutoff on the minimum ORF length, which lead to genes encoding proteins with less than 100 amino acids (referred to as microproteins) being annotated as lncRNA. Interestingly, several of such short open reading frames (ORFs) are found in lncRNAs, adding another exciting facet to the function of this highly diverse class of non-coding RNAs. Some of these microproteins have been characterized as important regulators of cancer [42,43,44,45]. Thus, microproteins could potentially be promising therapeutic targets as well. Their role in LDM and the development of therapy resistance has so far not been investigated. 

RNA-seq-based expression studies revealed that circRNAs are conserved and their expression patterns are highly tissue-specific [46,47,48,49]. circRNAs are also differentially expressed in cancer [50,51,52] and may even serve as biomarkers, since they are abundantly found in body fluids such as blood [53]. In addition, they appear to be rather stable due to the lack of free termini accessible for exoribonucleases. Recent studies reported comprehensive landscapes of circRNA expression in prostate cancer, as well as other tumor entities [54,55]. Strikingly, knockdown screening of a large panel of circRNAs revealed that more than 10% are important for cell proliferation while their parental, linear transcripts are not [54]. Thus, circRNAs comprise a class of non-coding RNAs with important functions in cancer.

In this study, we have investigated expression patterns of both coding and non-coding transcripts using a preclinical intrasplenic injection mouse model for colorectal cancer progression under LDM topotecan chemotherapy [19]. This model, although partly regarded as a homing model for circulating tumor cells, also serves as a suitable metastasis model for the experimental questions of the present study [19]. Our sequencing data reveal expression changes of several miRNAs upon treatment. Particularly, we found that both miR-210-3p and miR-210-5p, as well as their host gene *MIR210HG* are affected by LDM topotecan chemotherapy. Furthermore, we found that the expression of a group of interferon-induced genes was most upregulated and thus define a specific gene expression signature potentially important for LDM chemotherapy. Among several identified lncRNAs, *LINC01133* and *LINC00483* were upregulated during LDM treatment. While *LINC01133* has no coding potential, we show that *LINC00483* contains an ORF and may have coding potential, suggesting that microproteins could play a role during cancer progression and therapy response. Finally, we identified several circRNAs that change during colorectal cancer progression as well as during LDM treatment. We focused on characterization of circZNF609 and found that circZNF609 is upregulated during colorectal cancer progression and metastasis. Using in vitro and in vivo mouse xenograft experiments, we demonstrated that circZNF609 functions as oncogene and is important for colorectal cancer progression and liver metastasis.

## 2. Results

### 2.1. A mouse Model for Profiling Dysregulated Genes during LDM Topotecan Chemotherapy 

To find candidate coding and non-coding RNAs including miRNAs, lncRNA and circRNAs, and to unravel their molecular functions during colorectal cancer progression, we utilized a xenograft mouse model based on implantation of the HT29.hCG.Luc colorectal cancer cell line (referred to as HT29) [19]. HT29 cells were injected into the spleen and tumors developed after three to seven days. After one week, LDM topotecan chemotherapy was initiated in the treatment group, while the control group received sodium chloride accordingly (Figure 1A). LDM topotecan chemotherapy was stopped after six to eight weeks and tissue from the primary injection site as well as from resulting liver metastases was harvested for RNA-seq (Figure 1A and Appendix A). For a global view on gene expression changes, we performed miRNA and total rRNA-depleted RNA-seq from HT29 cells, primary tumors and liver metastases in control and treated mice (Figure 1A).

### 2.2. MiRNA-Seq Revealed miRNA Dysregulation during LDM Topotecan Chemotherapy

To unravel a potential role of miRNAs in response to LDM topotecan chemotherapy, we analyzed dysregulated miRNAs in the above-mentioned model. Principal component analysis revealed that replicates of HT29 cells (HT29-1-3), primary tumors from control mice (C-PT), liver metastases from control mice (C-LM) and liver metastases from treated mice (T-LM) cluster well together, suggesting common gene expression programs within these four groups (Figure 1B and Appendix A). Mean-log ratio (MA) plots show several miRNAs that are up- or downregulated in both C-PT (Figure 1C) and C-LM (Figure 1D) compared to HT29 samples, suggesting a dynamic remodeling of miRNA populations during cancer progression. To identify miRNAs relevant for LDM topotecan chemotherapy, we compared miRNA signatures of liver metastases subjected to topotecan treatment with control mice (Figure 1E,F and Appendix A, Appendix A). miR-210-3p, miR-200c-3p and miR-429-3p are known to be important for metastasis [56,57] and were thus selected for further validation by Northern blotting (Figure 1G). Although we also tested miR-1246, which was upregulated in liver metastases after treatment with topotecan chemotherapy in our sequencing data (Figure 1F), we did not succeed to detect miR-1246 by Northern blotting (personal conmmunication). Interestingly, not only were miR-210-3p and miR-210-5p (Figure 1F) significantly downregulated in LDM-treated samples, but their host gene MIR210HG was downregulated as well (Figure 2B and Figure 3A), indicating transcriptional downregulation of the three transcripts (Figure 1H, upper panel). Validation of the expression of MIR210HG using two independent primer pairs (P01 and P02) showed that MIR210HG is upregulated in both primary tumor and liver metastasis in control mice (C-PT and C-LM) and downregulated in treated liver metastases (T-LM) (Figure 1H, lower panel). This suggests that both MIR210HG as well as miR-210 are affected by LDM topotecan treatment and thus could be important during metastatic progression of colorectal cancer.

### 2.3. Protein Coding Gene Expression Analyses Identify Several Immune-Regulated Genes Affected by LDM Topotecan Treatment

As protein coding genes are still the main targets for cancer therapy, understanding the molecular changes of these mRNA under LDM topotecan chemotherapy may provide further improvements in cancer treatment. We therefore investigated mRNA expression signatures of the rRNA-depleted RNA-seq data from our mouse model described above. Principal component analysis from our RNA-seq data revealed consistent clustering as shown for miRNAs (Appendix A). We then focused on the analysis of the differentially expressed mRNA between treated and control liver metastases. A complete list of upregulated and downregulated genes in T-LM compared to C-LM samples can be found in Appendix A, while only the most significantly regulated genes are highlighted in Figure 2A,B. We found less genes that are downregulated in LM upon treatment with topotecan than upregulated genes. In agreement with previous reports, several genes including *NDRG1* [58] or *PFKFB3* [59,60] were also downregulated upon LDM treatment in our experiments. Some of these candidates were selected for further validation by qPCR (Figure 2C). qPCR analyses confirm that the selected candidates *BHLHE40*, *EFNA1, NDRG1, PFKFB3, ZBED9, FOS* and *SLC5A9* are downregulated in LM upon LDM topotecan treatment. *NDRG1* is the most significantly affected gene of the validated candidates. Interestingly, *NDRG1* was previously shown to be a metastasis suppressor and is highly downregulated in colorectal cancer [61,62]. 

Furthermore, we found several regulated genes belonging to the group of the ISGylation pathway, which is important for ISG15 conjugation [64,65]. For example, *ISG15, HERC5, USP18* and *UBE2L6* are upregulated in liver metastases upon LDM topotecan treatment (Figure 2A,B,D). Moreover, several interferon-induced genes such as *IFI6, IFI16, IFI27, IFI44, IFI44L, IFIT1, IFIT2, IFIT3, IFITM1, ASAD2, CXCL10, CXCL11* and *CCL22* were generally upregulated in liver metastasis upon LDM treatment (Figure 2A). We selected the candidates *IFI16, IFI44, OASL, IFITM1, ASAD2, HERC5, XAF1* for further validation by qPCR and we could successfully confirm their expression levels observed in the RNA-seq data (Figure 2D). 

One of the most significantly upregulated genes in our model is the poorly characterized Zymogen granule protein 16 (*ZG16*). *ZG16* is strictly expressed in the gastrointestinal tract [66] and is significantly downregulated in colorectal cancer samples as evident from the TCGA database (Figure 2E) [63,67]. Consistently, low expression of *ZG16* is associated with poor survival based on TCGA data (Figure 2F) [63]. Since *ZG16* is significantly upregulated in LM upon LDM topotecan treatment, *ZG16* might be an interesting target for cancer therapy (Figure 2D).

### 2.4. LncRNA Expression Analysis Identifies LINC01133 and LINC00483 as Potential LDM Topotecan Chemotherapy Targets

Regulatory lncRNAs play important roles both in cancer progression and response to chemotherapy [30,35,68]. We therefore analyzed lncRNA expression patterns in C-PT and C-LM from the above-mentioned mouse model. First, we performed Northern blotting and found that *NEAT1* short isoform1 (NEAT1 v1) was highly upregulated in both C-PT and C-LM in the non-treated group (Appendix A). *NEAT1* is a nuclear lncRNA that functions in paraspeckles [69] and tumorigenesis [39]. Furthermore, *MALAT1*, an abundant lncRNA that has been implicated in cancer metastasis [70], was slightly upregulated in C-LM (Appendix A). These positive controls encouraged us to further investigate other candidate lncRNA in our datasets. We selected candidates differentially regulated in C-PT and C-LM for validation by qPCR (Appendix A). Indeed, all candidates tested showed expression levels as observed in our RNA-seq data. 

Next, we analyzed lncRNAs differentially regulated during LDM topotecan treatment in our model system (Figure 3A). We found two interesting candidates, *LINC01133* and *LINC00483*, that are upregulated during topotecan treatment (Figure 2B and Figure 3B). Importantly, both *LINC01133* and *LINC00483* are downregulated in colorectal cancer from TCGA data (Figure 3C,D) and are associated with cancer survival (Figure 3E,F). Furthermore, when we analyzed the expression of these two lncRNAs in different colorectal cancer cells treated with topotecan in vitro, we found that both lncRNAs were significantly upregulated after two days treatment with topotecan in several tested cell lines (Appendix A). While *LINC01133* is well characterized as an important factor in colorectal cancer progression and metastasis [71], *LINC00483* is poorly characterized. Surprisingly, *LINC00483*, a gastrointestinal tract specific RNA (Appendix A), is highly conserved among vertebrates and potentially encodes for small proteins (Appendix A).

### 2.5. circZNF609 as Upregulated circRNA during Cancer Progression

Since many reports suggest that circRNAs can be relevant for cancer [55,72], they might be potent candidates for therapeutic strategies. We first focused on the analysis of circRNAs regulated during colorectal cancer progression and metastasis by analyzing the samples from HT29, C-PT and C-LM. For this purpose, we analyzed circRNA expression by detecting reads overlapping head-to-tail junctions, indicative of circular splicing. Interestingly, several circRNAs were differentially expressed, while their cognate mRNAs were largely unchanged (Figure 4A–C and Appendix A, Appendix A). To validate our RNAseq data, we selected a number of candidates and performed qPCR experiments using circRNA-specific primer pairs (Figure 4D). Most of the selected circRNAs were regulated in a similar way as observed in our RNA-seq data and only circTANK, circTNFRSF21, circZKSCAN1 and circUBAP2 showed a rather mild regulation. Thus, our validations confirm up- or downregulation of selected circRNA candidates in independent samples underscoring the reliability of the sequencing data (Figure 4D). Subcellular localization patterns of circRNAs are important for further functional characterization. We separated nuclear and cytoplasmic fractions from HT29 cells and determined circRNA levels in each fraction by qPCR (Figure 4E and Appendix A). Most candidates are predominantly found in the cytoplasmic fractions, while some (e.g., circCCDC9 or circTANK) are also present in the nucleus, suggesting efficient export and potential cytoplasmic functions of these circRNA candidates.

For further functional and mechanistic characterization, we selected circZNF609, which is generated from the second exon of the *ZNF609* pre-mRNA (Figure 4F). It is upregulated during cancer progression and metastasis (Figure 4D), absent in poly(A)-selected RNA fractions (Appendix A) and resistant to RNase R treatment, a 3′ to 5′ exoribonuclease that generally degrades linear RNAs while circRNAs remain unaffected (Figure 4F,G). In addition, northern blotting detected circZNF609 as a strong band migrating at the predicted size, confirming robust expression as well as the circular nature of this molecule (Figure 4F). Moreover, circZNF609 is highly conserved between human and mouse, suggesting that our mouse model can be utilized to recapitulate mechanisms relevant for human colorectal cancer progression and treatment (Appendix A). Interestingly, circZNF609 has recently been associated with cell cycle progression in rhabdomyosarcoma [73] and we thus concluded that this circRNA could be important in colorectal cancer as well.

### 2.6. CircZNF609 Promotes Colorectal Cancer Progression in Mouse Xenografts

The increased expression levels of circZNF609 in primary tumors as well as liver metastases suggest a potential role in colorectal cancer. To investigate the impact of circZNF609 on colorectal cancer development in vivo, we generated cell lines allowing for an inducible knockdown of circZNF609 (Appendix A). We generated Doxycycline (Dox)-inducible HEK293T (Figure 5A), HCT116 (human), HT29 (human) (Figure 5B) and CMT93 (mouse) (Figure 5F) cells expressing a shRNA-derived siRNA targeting the back-spliced junction of circZNF609 (Figure 5A and Appendix A). Of note, these cell lines also contain a stably integrated luciferase gene for monitoring in xenografts. Doxycycline addition indeed results in increased expression of the desired siRNAs, which is efficiently bound to Ago2 and therefore functional (Appendix A). Doxycycline induction leads to a strong decrease in circZNF609 levels but the parental ZNF609 mRNA remains unchanged, indicating that our knockdowns are specific and the cell lines can be used for further functional studies (Figure 5A,B,F).

We first investigated effects of circZNF609 knockdown on cell proliferation and performed XTT assays (Figure 5C). We found that knockdown of circZNF609 upon doxycycline induction decreased cell proliferation of both HT29-sh-circZNF609 and HCT116-sh-circZNF609 cells. To further corroborate these in vitro observations, we tested effects of circZNF609 knockdown in vivo. 500,000 HT29-sh-circZNF609 cells were injected into the spleen of SCID mice. Depending on the luciferase signals, mice were sacrificed accordingly. Spleens containing the primary tumor as well as livers containing metastases were resected and the tumor mass was determined (Figure 5D,E). Upon doxycycline induction, less primary tumor as well as fewer and smaller liver metastases were found compared to the control group (Figure 5D,E). RNA from tumors and metastases were collected and qPCR analyses were performed. Consistently with our in vitro results, a significant decrease in the expression of circZNF609 was observed while the expression of *ZNF609* remained rather unchanged (Figure 5D,E and Appendix A).

We further tested whether our observations are conserved in mice and validated the effect of circZNF609 knockdown in a murine CMT93 cell line (Figure 5F). To further strengthen our findings, we used an independent xenograft model system. 500,000 CMT93-sh-circZfp609 cells were subcutaneously injected into C57BL/6 mice. Although statistically not significant due to the low number of tumors developed under doxycycline induction, tumor weight was generally lower (Figure 5G and Appendix A), which is consistent with the data obtained from the HT29-sh-circZNF609/SCID mouse model (Figure 5D,E). 

In addition to the knockdown studies and to further confirm a potential oncogenic role of circZNF609, we tested the effect of circZNF609 overexpression in vivo. We generated an inducible stable cell line CMT93 expressing circZfp609 (CMT93-circZfp609) using overexpression system via ZKSCAN1 flanking introns described earlier [74,75,76] upon doxycycline induction (Figure 5H). As expected, northern blotting showed a strong increase of circZfp609 levels upon doxycycline treatment (Figure 5H). Next, we injected 500,000 cells of the CMT93-circZfp609 stable cell line subcutaneously into mice. Upon induction with doxycycline, mice developed bigger tumors compared to the non-induced control mice (Figure 5I and Appendix A). Thus, our overexpression studies are consistent with our knockdown experiments and we therefore conclude that the circZNF609 might function as oncogene to promote metastatic colorectal cancer progression.

## 3. Discussion

In this study, we followed up on our earlier study showing that LDM topotecan chemotherapy has profound effects in decreasing liver metastasis and increasing mouse survival [19] in colorectal cancer. Using RNA sequencing, we identified several genes that are differentially expressed during LDM treatment and investigated the potential of these molecules as regulators during colorectal cancer progression. 

We found that miR-210-3p and miR-210-5p are both downregulated after treatment with LDM topotecan chemotherapy in our mouse model suggesting that these miRNAs are affected by LDM topotecan treatment. MiR-210 is a hypoxia-induced miRNA [77,78] and associated with radio-resistance in lung cancer [79]. Moreover, miR-210-3p was shown to be downregulated in colorectal cancer cells treated with 5-fluorouracil (5-FU), leading to an adaptation of cancer cells during treatment and enhancing chemotherapy resistance [80]. While the role of miR-210 is well studied during hypoxia, cancer progression and therapy, potential miRNA-independent functions of its host gene *MIR210HG* are less studied. Similar to miR-210, *MIR210HG* has been identified as a highly upregulated lncRNA in endothelial cell response to hypoxia [81]. Moreover, *MIR210HG* was identified as prognostic marker of colorectal cancer [82] and shown to promote the proliferation and invasion of non-small-cell lung carcinoma via inhibiting the expression of *CACNA2D2* by recruiting *DNMT1* [83]. Yet, the function of *MIR210HG* has not been reported during chemotherapy. In this study, we found that *MIR210HG* was downregulated after LDM treatment with topotecan suggesting that it is a potential target of LDM chemotherapy. Since miR-210-3p and miR-210-5p are generated from an intron of MIR210HG, all these three transcripts are probably produced from the same promoter. Previously, it was shown that the transcription factor *HIF-1*, an important anticancer therapeutics target [84], binds to the above mentioned promoter of miR-210 and regulates its expression during hypoxia [78]. It will therefore be interesting to investigate, if these potential downstream targets of *HIF-1*, miR-210-3p, miR-210-5p and *MIR210HG*, work together or use distinct modes of action under treatment with topotecan. 

Protein coding genes are still the most important therapeutic targets of cancer therapy. In our study, we identified a signature of upregulated transcripts during LDM treatment, which belongs to the family of interferon-stimulated genes. Moreover, using an in vitro cell-based therapy resistance model, we found that some of these genes including *XAF1, IFITM1, OASL, IFIT44*, were upregulated in HT29-resistant cell lines suggesting that they could be mediators of chemotherapy resistance (Appendix A). Since cytokines are important molecules for cancer cells allowing them to evade anti-tumor immune responses [85], the dysregulation of interferon-induced genes in our LDM treatment model highlights the complexity of the tumor and its microenvironment under the pressure of chemotherapy treatment. Our findings suggest that cancer cells might take advantage of the interferon signaling pathway for modulation of LDM topotecan chemotherapy resistance. Moreover, we also found that the ISGylation pathways may contribute to resistance during LDM topotecan chemotherapy. For example, we found that *ISG15* was upregulated after treatment with LDM topotecan. *ISG15* was previously shown to be a novel target for camptothecin sensitivity [86]. Another upregulated protein of this pathway is *HERC5*, a HECT-type E3 protein ligase, which mediates ISGylation of protein targets [87]. Consistent with previous findings in topotecan-treated and -resistant ovarian cancer cells [21], we found that *HERC5* was upregulated in LM after LDM topotecan treatment and in HT29-topotecan resistant cell lines (Appendix A). This implies that *HERC5* may serve as important regulator during the development of cancer therapy resistance. 

LncRNAs have been reported to be dysregulated in all types of cancer [30,31]. However, the detailed function of lncRNAs and their molecular mechanisms are often not fully clear. For example, the most abundant and well characterized lncRNA *MALAT1* has been characterized as important driver of lung metastasis [70]. Importantly, it has also been reported that *MALAT1* is a suppressor of breast cancer metastasis [88,89]. Therefore, a better understanding of lncRNA is needed in order to develop therapeutic applications targeting lncRNAs.

In depth computational but also transcriptome-wide ribosome binding studies revealed that several lncRNAs contain short ORFs that are actively translated [90] (now commonly referred to as microproteins [41]). Therefore, these “lncRNAs” are actually coding. In our study, we found that *LINC00483* was upregulated upon treatment with LDM topotecan. Interestingly, *LINC00483* was previously identified as lncRNA that promotes gastric cancer proliferation [91] as well as colorectal cancer proliferation and metastasis [92]. Furthermore*, LINC00483* produces several transcripts, some of which indeed may encode for small peptides (Appendix A). Indeed, it was recently re-annotated as *ANKRD40CL* protein coding gene and its protein product is highly similar to the C-terminus of *ANKDR40* (Appendix A). The molecular functions of this short peptide, however, remain elusive. Since L*INC00483* is significantly downregulated in colorectal cancer, it would be interesting to investigate the impact of the lncRNA *LINC00483* as well as its potential protein products for successful chemotherapy treatment. 

Although the individual functions of circRNAs as regulatory RNAs are poorly understood, circRNAs have been indicated to be expressed in a tissue-specific manner [47,48,49,93] and associated with several types of cancer [52,72]. In the above-mentioned colorectal cancer mouse model, we found that circZNF609 was upregulated in primary tumors as well as liver metastases and identified circZNF609 as a potential regulator important for metastatic colorectal cancer progression. Inhibition and overexpression resulted in significant effects on tumor size and weight. Moreover, circZNF609 was downregulated after LDM topotecan treatment in our mouse model (Appendix A), suggesting an important role during chemotherapy. Interestingly, several circRNAs contain ORFs, and it has been shown that circRNAs can be translated [94,95]. These proteins have been reported in cancer [44] even though their precise roles are largely unknown. CircZNF609 contains an ORF that generates a protein resembling a short fragment of the full-length parental transcription factor ZNF609, which has been implicated in the regulation of neuron migration [96]. However, an endogenous peptide produced from circZNF609 is very difficult to identify and either not present or very low abundant. We therefore concluded that most of the observed circZNF609 protein products likely originate from trans-splicing by-products of the overexpression vector and that circZNF609 functions as non-coding RNA [74]. Nevertheless, circZNF609 is upregulated in rhabdomyosarcoma and it regulates cell cycle progression [73]. It will be interesting to see whether cell cycle control is also affected by circZNF609 in colorectal cancer cells and whether this is a general function of this regulatory RNA probably even in other types of cancer. In agreement with this hypothesis, circZNF609 was recently implicated in colorectal cancer [97] and breast cancer [98]. Of note, downregulation of circZNF609 was observed in colorectal cancer as well [99]. It is therefore important to carefully examine the expression of circZNF609 in orthotopic intracecal mouse models of spontaneous liver metastases as well as in patient-derived colorectal cancer tissues. For example, by analyzing the spatial expression, the oncogenic ciRS-7 was recently reported to be absent in colon cancer cells while expressed at high levels in stroma cells [100]. Similarly, circZNF609 was also found to be expressed higher in surrounding stroma cells compared to colon cancer cells [100]. Interestingly, we found that human circZNF609 is circulating in the serum and blood of SCID mice using targeted deep sequencing of the amplicon spanning junction of circZNF609 (personal communication). Even though we found that circZNF609 potentially functions as oncogene to promote colorectal cancer progression, the underlying molecular mechanisms still need to be unraveled. We searched for potential miRNA binding sites within the circZNF609 sequence by computational prediction [101] and found 41 miRNAs having one binding site to circZNF609 and four miRNAs having two binding sites to circZNF609 (Appendix A). Of these 45 miRNAs, 20 miRNAs were not found in our miRNA-seq datasets. From the 25 miRNAs that are present in our datasets, 18 miRNAs appear to be very low abundant and thus a sponging function of circZNF609 for these miRNAs is most likely not physiologically relevant. We still do not know if the remaining seven miRNAs could contribute to function of circZNF609 and will need further investigation. 

## 4. Materials and Methods

### 4.1. Plasmids

pMD2.G (Addgene plasmid #12259) and psPAX2 (Addgene plasmid #12260) were a gift from Didier Trono (EPFL, Lausanne, Switzerland). TRMPV-Hygro (Addgene plasmid #27996) was a gift from Scott Lowe (Sloan Kettering Institute, Memorial Sloan Kettering Cancer Center, New York, NY, USA). pLVX-TetOne-Puro (Clonetech, A Takara Bio Company, Shimogyō-ku, Japan) and TRMPV-Hygro were used as backbone to generate pLVX-TetOne-Venus-IRES-HygB plasmid. pcDNA3.1(+) ZKSCAN1 MCS-WT Split GFP (Addgene plasmid #69908) was a gift from Jeremy Wilusz (University of Pennsylvania Perelman School of Medicine, Philadelphia, PA, USA). pcDNA3.1(+) ZKSCAN1 MCS-WT Split GFP was used as backbone to clone pcDNA3.1(+) ZKSCAN1-circZNF609 as described earlier [74]. pSuperior-puro-GFP (oligoengine) was used as backbone to generate pSuperior-Zeocin-GFP plasmid. pLVX-TetOne-Venus-IRES-HygB was used to generate pLVX-TetOne-Venus-IRES-HygB-sh-circZNF609. All the sequences of the oligos can be found at Appendix A.

### 4.2. Antibodies

The following antibodies were used for western blots: rat-anti-Ago2 (monoclonal, clone 11A9 [102], hybridoma supernatant, 1:10), rabbit-anti-GAPDH (polyclonal, clone FL-335, 1:500, Santa Cruz Biotechnology, Heidelberg Germany), mouse-anti-p54nrb (NONO) (monoclonal, clone 3, 1:1000, BD Biosciences, Franklin Lakes, NJ, USA). Goat-anti-rabbit IRDye 680RD or goat-anti-mouse/rat/ IRDye 800CW antibodies (Li-Cor Biosciences, Lincoln, NE, USA) were used as secondary antibodies.

### 4.3. Cell Lines, Cell Culture and Generation of Stable Cell Lines

HEK293T, HCT116.hCG.Luc, HT29.hCG.Luc, CMT93.hCG.Luc, SW620, Caco-2 cells were cultivated in Dulbecco’s modified Eagle’s medium (DMEM) supplemented with 10% fetal bovine serum (FBS) and penicillin/streptomycin (P/S) antibiotics mix (all from Sigma-Aldrich, St. Louis, Missouri, USA). HCT116.hCG.Luc, HT29.hCG.Luc, CMT93.hCG.Luc were generated as previously described [19]. HT29-topotecan resistant cell line was generated by exposing the HT29 parental cells to the increasing amount of topotecan for six months (from the initial amount of topotecan treatment 1nM to 30 nM of topotecan after six months). For generation of stable HEK293T, CMT93.hCG.Luc cell line expressing desired constructs (see Appendix A), the cells were transfected with plasmids in 6-well format. Cells were split one day post-transfection into one 10 cm plate and selection was started 2 days post-transfection with 100 µg/mL Zeocin (Thermo Fisher Scientific, Waltham, MA, USA). The cells were then sorted for clonal GFP-expressing cells using FACSAria II cell sorter (BD Biosciences, Franklin Lakes, NJ, USA). Stable clones were maintained in the same medium as used for selection. Lentiviral transduction was used for generation of HCT116.hCG.Luc and HT29.hCG.Luc stable cell lines. Briefly, lentiviral packaging was performed in HEK293T cells as previously described using 5 μg pMD2G, 10 μg psPAX2, 20 μg lentiviral plasmids pLVX-TetOne-Venus-IRES-HygB-sh-circZNF609. The HT29.hCG.Luc, HCT116.hCG.Luc cells were plated and propagated in 6 well-plates overnight. On the next day, after changing to the new medium, virus was added to the cells. Two days later, transduced cells were selected by 300 µg/mL Hygromycin B (Invitrogene, Thermo Fisher Scientific, Waltham, MA, USA) for 3 passages. Afterward, the cells were sorted for clonal GFP-expressing cells using FACSAria II cell sorter (BD Biosciences, Franklin Lakes, NJ, USA). Stable clones were maintained in the same medium as used for selection.

### 4.4. Mouse Strains 

The following mouse strains were used or generated during this study: Both C57BL/6 and SCID were obtained from Charles River Laboratories (Sulzfeld, Germany).

### 4.5. Intrasplenic Implantation

Five SCID mice were initially used for tumor development to collect RNA for RNA sequencing (RNA-seq) via intrasplenic implantation. HT29.hCG.Luc cells were dissociated with trypsin, washed once with DMEM supplemented with 10% FCS, 1% PS and then resuspended to the desired concentration in Hank’s Balanced Salt Solution (HBSS, Sigma-Aldrich, St. Louis, MO, USA). All animals receiving tumor cell injections were injected with the same quantity of cells (500,000 cells per animal).

Abdominal access was obtained via a left subcostal skin and peritoneal wall incision. The spleen was gently exteriorized. Five µL containing 500,000 tumor cells were injected with a 10 µL Hamilton syringe (BD Biosciences, Franklin Lakes, NJ, USA). The needle was slowly retracted and the injection site pressed with a moist cotton swab to prevent leakage. The spleen was returned to the peritoneal cavity; peritoneum and skin were closed by running sutures and wound clips.

Eight SCID mice were used to validate the effect of circZNF609 knockdown on tumor development in vivo using the stable cell lines HT29.hCG.Luc-sh-circZNF609.

### 4.6. Subcutaneous Implantation

Ten C57BL/6 mice were used to test the effect of circZfp609 knockdown and circZfp609 overexpression on tumor development in vivo using the stable cell lines CMT93.hCG.Luc-sh-circZfp609 and CMT93.hCG.Luc-circZfp609, respectively. 50 µL of 500,000 tumor cells were injected subcutaneously into C57BL/6 mice using a manual 1 mL Hamilton syringe and 30G needle (BD Biosciences, Franklin Lakes, NJ, USA). Tumor size was measured daily starting five days after implantation with calipers and tumor volume calculated as width × width × length.

### 4.7. Animal Handling

Bioluminescence measurement was performed weekly or every three days. For this purpose, mice were anaesthetized, injected with 100 µL of 30 mg/mL D-luciferin (Biosynth, Staad, Switzerland) for three minutes, and emitted photons were registered for three minutes using Xenogen IVIS Lumina Imaging System (Caliper Life Sciences, A PerkinElmer Company, Hopkinton, MA, USA).

Doxycycline was obtained from Sigma-Aldrich, St. Louis, MO, USA. The mice were gavaged daily with 100 µL of 5 mg/mL Doxycycline as described [103].

Primary tumor (either subcutaneous tumor or intrasplenic) and hepatic metastases resections were performed when the tumor size was big enough or when the bioluminescence measurement was significant. Samples of primary tumors and liver metastases were weighted and imaged with a normal camera. All tumors were divided for RNA (collected in RNAlater (Thermo Fisher Scientific, Waltham, MA, USA)), protein and immunohistochemistry analyses.

### 4.8. Preparation of Small RNA Library and Data Analysis

The small RNA library was prepared from total RNA. Isolated RNA was ligated to an adenylated 3′ adapter by a truncated T4 RNA Ligase 2 [104] (expressed and purified from Meister lab), the 5′ RNA adapter was added in a second ligation step by T4 RNA Ligase 1 (NEB, Ipswich, MA, USA). The product was reverse-transcribed using the SuperScriptIII First Strand Synthesis Super Mix (Invitrogen, Thermo Fisher Scientific, Waltham, MA, USA) using a specific primer, followed by a PCR amplification. The samples were run on a 6% Urea-PAGE, the bands corresponding to small RNA containing ligation products were cut out and eluted overnight in elution buffer (300 mM NaCl, 2 mM EDTA). The libraries were precipitated with ethanol overnight at −20 °C, then collected by centrifugation and solved in water. 

The library was sequenced on a MiSeq (Illumina, San Diego, CA, USA) in a 1 × 50 bp run. Sequences were counted against human miRNAs listed in the miRBase v22.1 (December 2018; http://www.mirbase.org) using mirDeep2 [105] with standard parameters. The minimum length of reads was set to 18 nucleotides. Annotated miRNA reads were normalized as reads per million mapped reads (RPM) values according to the total number of mapped reads in the respective library. Differential expression analysis was performed within R [106] using DESeq2 version 1.24.0 [107]. 

### 4.9. Preparation of rRNA-Depleted Library and Analysis of mRNAs and lncRNAs

The rRNA-depleted RNA library was prepared using the Ovation Human FFPE RNA-Seq Library System (NuGEN, TECAN, Männedorf, Zürich, Switzerland) according to manufacturer’s protocol. Specifically, 100 ng of total RNA was used. The Sequencing was performed on an Illumina HiSeq 1000 Sequencing Platform with Base calling using CASAVA 1.8.2 software (Illumina, San Diego, CA, USA).

For mRNA and lncRNA analysis, raw reads were mapped to the human genome (hg38 assembly) using STAR version 2.7.0a [108] with --outFilterMultimapNmax 2000000000 and --outSAMmultNmax 1 settings. Reads were counted using Rsubread 1.34.7 with Ensembl release 98 genome annotation reference. Differential expression analysis was performed within R using DESeq2 version 1.24.0.

### 4.10. circRNA Detection and Differential Expression Analysis

For circRNA analysis, raw reads were mapped to the human genome (hg19 assembly) using STAR version 2.5.3a [108]. We used the default settings, with the exception of --outFilterMismatchNoverLmax, which was set to 0.05. Reads were counted using htseq-count (version 0.9.1 [109]) with Ensembl release 75 genome annotation reference [110]. 

Reads were mapped to the human (hg19) genome using bwa mem (version 0.7.5a-r405) with the following options: -t 20 -k 14 -T 1 -L 3,3 -O 6,6 -E 3,3. bwa mem output was processed using find_circ2 (https://github.com/rajewsky-lab/find_circ2) to detect backspliced reads. circRNA candidates flagged by find_circ2 as SHORT or HUGE were discarded. The list was further filtered by keeping only circRNA candidates that were supported by 2 or more unique RNA-seq reads in a given sample, and spliced from annotated splice sites. Number of fragments overlapping head-to-tail junction was used as a measure of circRNA expression.

Differential expression analysis was performed using the DESeq2 (version 1.22.2) R package [107], with default options. circRNA head-to-tail junction counts were merged with htseq-count output prior to normalization by DESeq2. circRNAs and genes that were supported by less than 10 reads in total across all samples were discarded. Adjusted p-value of 0.05 was used as the significance cutoff.

### 4.11. Validation of circRNAs Expression by qPCR

Sequences of selected differentially expressed circRNAs were obtained from circBase [111] and CircInteractome [101]. Designed divergent qPCR oligos are listed in Appendix A. For qPCR, RNA was preferably isolated using the NucleoSpin RNA Kit (Macherey-Nagel, Düren, Germany). Otherwise, if RNA had been isolated by TRIzol (Life Technologies, Thermo Fisher Scientific, Waltham, MA, USA), the RNA was then treated on column using DNA digestion step from NucleoSpin RNA Kit (Macherey-Nagel, Düren, Germany). One µg RNA was used for cDNA synthesis using First Strand cDNA Synthesis Kit (Thermo Fisher Scientific, Waltham, MA, USA). qPCR was done with Sso Fast Eva Green Mix (Bio-Rad, Hercules, CA, USA) using 0.5 µM forward and 0.5 µM reverse primer and cDNA from 10 ng RNA as template. qPCR was run on a CFX96 cycler (Bio-Rad) using the standard program as given in the SsoFast EvaGreen SuperMix manual. Data were evaluated using the ∆∆Ct method with GAPDH or RPL32 as reference mRNA for normalization.

### 4.12. Proliferation Assay 

Single cell suspensions were obtained after trypsinization and passing through a cell strainer. XXT were performed in 96 well plates using XTT colorimetric assay kit (11465015001, Roche, Switzerland) based on manufacturers instruction. Specifically, 300 cells/well for HT29.hCG.Luc or HCT116.hCG.Luc cell lines or 100 cells/well for CMT93.hCG.Luc cell lines were seeded in 10 replications. 6 h after seeding, 5 replications were added with DMEM media supplemented with 10% FCS, 1% PS while the other 5 replications were added with DMEM media supplemented with 10% FCS, 1% PS, 1 µg/mL doxycycline. The XTT were assayed every day for 6 days.

### 4.13. Nuclear and Cytoplasmic Fractionation

Nucleo-cytoplasmic fractionations were prepared according to [112] with slight modifications. HT29 cells were cultured up to 80% confluency, washed with PBS and harvested by trypsin. The cells were then passed through a cell strainer (BD Biosciences, Franklin Lakes, NJ, USA) and washed again with ice-cold PBS. After centrifugation at 100 g for 5 min at 4 °C, the cell pellet was resuspended by gentle pipetting with ice-cold hypotonic lysis buffer HLB (10 mM Tris (pH 7.5), 10 mM NaCl, 3 mM MgCl_2_, 0.3% NP-40 and 10% glycerol) and incubated for 8 min on ice. 1ml of HLB was used for every 75 mg cell pellet (or 10 million cells). Afterwards, cells were centrifuged at 800 g for 8 min at 4 °C. The supernatant of this step served as cytoplasmic fraction. The nuclei pellet was washed gently for 4 times with HLB through pipetting and centrifugation at 200 g for 2 min at 4 °C. Nuclei were resuspended in nuclear lysis buffer NLB (20 mM Tris (pH 7.5), 150 mM KCl, 3 mM MgCl_2_, 0.3% NP-40 and 10% glycerol). 0.5 mL of NLB was used for every 75 mg of initial cells used (or 10 million cells). If needed, the nuclear pellet was sonicated on ice one time with 10–20% power for 15 s. Both fractions were cleared by centrifuge at 15,000 g for 15 min at 4 °C. For RNA analysis, 250 µL of different fractions was collected in 750 uL of TRIzol-LS (Life Technologies, Thermo Fisher Scientific, Waltham, MA, USA) and isolated. 1 µg RNA from each fraction was used for quantification by qPCR.

### 4.14. SDS-PAGE and Western Blotting

For performing a western blot, samples were mixed with 5× Laemmli Sample Buffer, shortly heated at 95 °C for 5 min and then loaded onto a 10–15% polyacrylamide gel. After separation, the proteins were blotted onto an Amersham Protran Premium 0.45 µm membrane (GE Healthcare, Chicago, IL, USA) using 1× Towbin buffer for 1 min/kDa constant at 2 mA/cm^2^. The membrane was blocked in TBS containing 0.1% Tween-20 and 5% milk. After incubation with suitable antibodies, the membrane was scanned on a Li-Cor reader (Li-Cor Biosciences, Lincoln, NE, USA).

### 4.15. Preparation of RNA

RNA extraction from mammalian cells were performed with TRIzol reagent (Life Technologies, Thermo Fisher Scientific, Waltham, Massachusetts, USA) or NucleoSpin RNA Kit (Macherey-Nagel, Düren, Germany) following the manufacturer’s protocol. To isolate RNA from tissues, mouse tissue was mechanically disrupted using FastPrep-24 (MP Biomedicals, Santa Ana, CA, USA) with lysing matrix D containing 1mL TRIzol reagent. The program was set for 45 s at 6.5 m/s. Afterwards, the ceramic beads in TRIzol were incubated at RT for 5 min and then following the TRIzol manual. RNA extraction from the cell lysates was collected in TRIzol-LS reagent (Life Technologies, Thermo Fisher Scientific, Waltham, MA, USA) following the TRIzol-LS manual.

RNA treatment with RNase R was performed using RNase R (Biozym Hessisch Oldendorf, Germany) according to the manufacturer’s protocol. The reaction was cleaned up using NucleoSpin RNA Kit (Macherey-Nagel, Düren, Germany).

### 4.16. Northern Blotting

For detection of NEAT1, MALAT1 and circZNF609, 15–25 μg total RNA or RNA after treatment with RNAase R was separated on a 1% MOPS/Agarose gel. The detailed northern blotting protocol for MOPS/Agarose gel was described earlier [74]. After transferring the RNA onto Amersham Hybond-N membrane (GE Healthcare) via capillary botting, the RNA was UV-crosslinked to the membrane at 254 nm. For detection of miRNA, the RNA was loaded on a 12% polyacrylamide (acrylamid/bisacrylamid 19:1) urea gel (National Diagnostics, Atlanta, GA, USA) using 1× TBE buffer. Synthetic RNA with a length of 19, 21 and 24 nt was labeled with γ 32P-ATP (Hartmann Analytics, Braunschweig, Germany) and served as markers. The RNA was semi-blotted for 30 min at 20 V onto an Amersham Hybond-N membrane (GE Healthcare) and crosslinked to the membrane for 1 h at 50 °C using EDC solution (0.16 M 1-ethyl-3-(3-dimethylaminopropyl)carbodiimide, 0.13 M 1-methylimidazole pH 8.0) method as described [113].

The membrane was prehybridized at 50 °C–60 °C with hybridization solution (5× SSC, 20 mM NaPi pH 7.2, 7% SDS, 0.02% Albumin fraction V, 0.02% Ficoll 400, 0.02% polyvinylpyrrolidone K30). The short DNA probes were labelled with 20 µCi of γ 32P-ATP (Hartmann Analytics, Braunschweig, Germany) using T4 PNK enzyme (Life Technologies) according to the manufacturer’s protocol. The cDNA probes were prepared using Megaprime DNA-Labeling Systems kit (Cytiva, Global Life Sciences Solutions, Marlborough, MA, USA). The DNA or cDNA probes were purified using Illustra MicroSpin G-25 columns (GE Healthcare) and flow through were collected and used for hybridization.

The membrane hybridized with DNA probes was washed twice with wash solution 1 (5× SSC, 1% SDS), and once with wash solution 2 (1× SSC, 1% SDS) (10 min each wash). The membrane hybridized with cDNA probes was washed once with wash solution 3 (2× SSC, 0.1% SDS), once with wash solution 4 (0.5× SSC, 0.1% SDS), once with wash solution 5 (0.1× SSC, 0.1% SDS) (30 min each wash). After the last step, the membrane was wrapped in saran for exposure to a phosphor screen and then signals were scanned by Personal Molecular Imager System (Bio-Rad).

For stripping of the hybridized oligonucleotide, the membrane was washed twice with a boiling water supplemented with 0.1% SDS for 10 min. After the last washing step with boiling water for 10 min, the membrane was wrapped in saran and kept at −20 °C.

### 4.17. Ago2 Immunoprecipitations (IP)

Ago2 IP was performed as described previously [114]. Briefly, monoclonal antibodies for human Ago2 (11A9) [102] were coupled to Protein G Sepharose beads over night at 4 °C. Antibody-coupled beads were washed twice with PBS and once with lysis buffer (25 mM Tris-HCl pH 7.4, 150 mM KCl, 0.5% NP-40, 2 mM EDTA, 1 mM NaF). The lysate was added to the washed beads and incubated for 3 h of rotation at 4 °C. The beads were washed four times with wash buffer [300 mM NaCl, 50 mM Tris-HCl pH 7.4, 1 mM MgCl _2_, and 0.1% NP-40]. After the last wash, the beads were transferred to a new eppendorf tube and washed once with PBS.

For RNA extraction, the immunoprecipitates were treated with Proteinase K 0.2 mg/mL final concentration (40 µg per sample, in 200 µL 300 mM NaCl, 25 mM EDTA, 2% SDS, 200 mM Tris pH 7.5) at 55 °C for 30 min. The RNA was extracted with 200 µL of phenol:chloroform:isoamyl alcohol 25:24:1, pH 7.5–8.0 (Carl Roth, Karlsruhe, Germany) and precipitated with 2.5 volumes pure ethanol at −20 °C overnight. For input samples, 100 µL of the respective lysate was isolated using TRIZol reagents (Life Technologies, Thermo Fisher Scientific, Waltham, MA, USA) following the manufacturer’s protocol. The RNA was subsequently extracted according to the manufacturer’s instructions. 

### 4.18. Statistical Analyses 

Statistical analyses have been performed using GraphPad Prism (GraphPad Software, San Diego, CA, USA). *p* values less than 0.05 were considered to be significant. * *p* ≤ 0.05, ** *p* ≤ 0.01, *** *p* ≤ 0.001, ns: *p* > 0.05. Graphs and error bars reflect means ± s.d. (standard deviation).

## 5. Conclusions

In this study, we used a colorectal cancer mouse model and established gene expression profiles in primary tumors as well as liver metastases after LDM chemotherapy. We identified coding and non-coding transcript signatures and present highly promising miRNA, lncRNA and circRNA candidates, potentially relevant for future therapeutic applications. For a more detailed understanding of these molecules, future endeavors should focus on additional model systems such as an orthotopic model for colorectal cancer under LDM topotecan chemotherapy as described earlier [19]. Moreover, a single-cell based analysis of the population of recurrent tumor cells-driven chemotherapy resistance will help to understand the complexity of tumor-microenvironment interaction during LDM chemotherapy. Finally, results need to be validated and further investigated in human tumor samples. 

## Figures and Tables

**Figure 1 cancers-13-00049-f001:**
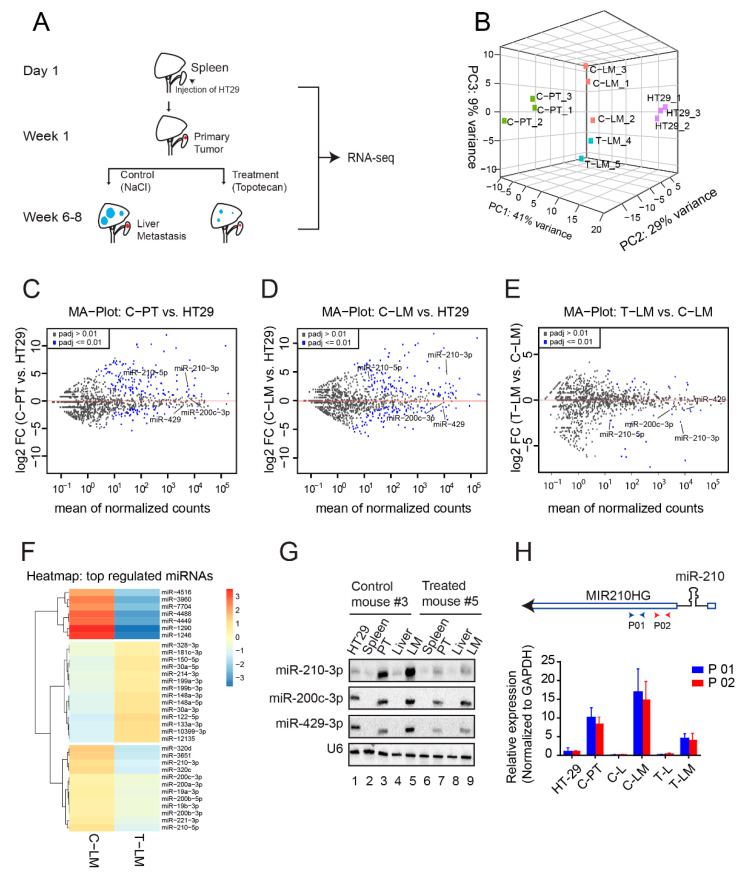
Mouse model and identification of miRNAs and *MIR210HG* as targets of LDM topotecan chemotherapy. (**A**) Schematic overview of the RNA collection procedure for library preparation. HT29 was intrasplenically injected into six weeks-aged female severe combined immunodeficient (SCID) mice for development of primary tumors. After one week, treatment with LDM topotecan chemotherapy was initiated, while the control group received sodium chloride. All mice were sacrificed when bioluminescence levels reached about 500,000 photons/s (about six to eight weeks after injection). The primary tumors from the spleens of control mice (C-PT) and metastases from livers of control mice (C-LM) and treated mice (T-LM) were collected for RNA sequencing. RNA from HT29 cells was collected at the time of injection. (**B**) Principal component analysis of the samples from miRNA library. (**C**–**E**) Mean-log ratio (MA) plot of miRNA expression in primary tumors from control mice (C-PT) versus HT29, liver metastasis from control mice (C-LM) versus HT29, and liver metastases from treated mice (T-LM) versus liver metastases from control mice (C-LM), respectively. (**F**) Heatmap of differentially expressed miRNAs in liver metastases from treated mice (T-LM) versus liver metastases from control mice (C-LM). (**G**) Northern blot validation of differentially expressed miRNAs, including miR-210-3p, miR-200c-3p, miR-429. U6 serves as loading control. PT (primary tumors), LM (liver metastasis). (**H**) Upper panel: Schematic representation of miR-210 and its host gene *MIR210HG*. P01 and P02 represents different oligoes for the detection of *MIR210HG* by qPCR. Lower panel: qPCR validation of MIR210HG using two different oligoes. C-PT: primary tumors from control mouse, C-L: liver from control mouse, C-LM: liver metastasis from control mouse, T-L: liver from LDM topotecan treated mouse, T-LM: liver metastasis from LDM topotecan treated mouse.

**Figure 2 cancers-13-00049-f002:**
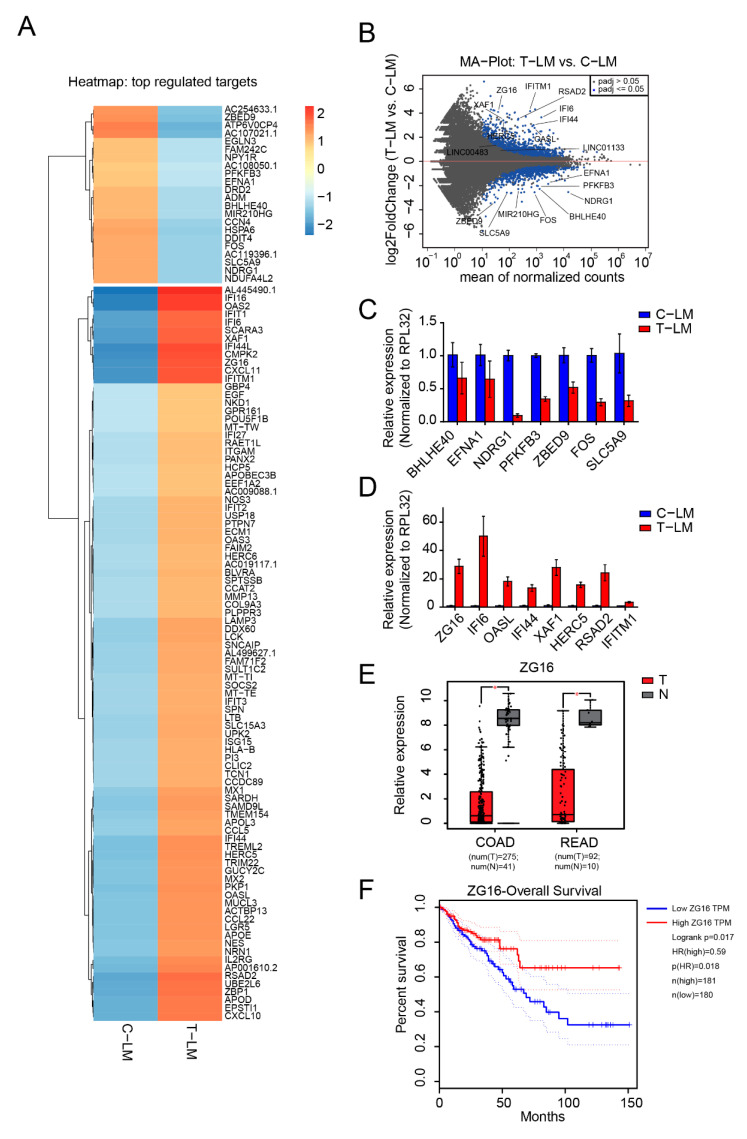
Identification of mRNAs regulated during LDM topotecan chemotherapy**. (A**) Heatmap of differentially expressed mRNAs in liver metastases from treated mice (T-LM) versus liver metastases from control mice (C-LM). (**B**) MA plot of mRNA expression in liver metastases from treated mice (T-LM) versus liver metastases from control mice (C-LM). (**C**,**D**) Validation of downregulated and upregulated mRNA from treated liver metastases compared to control liver metastases by qPCR. (**E**) Boxplot showing the significant downregulation of *ZG16* in colon adenocarcinoma (COAD) and rectum adenocarcinoma (READ) obtained from TCGA data, T: Tumors, N: Normal tissues. (**F**) Survival plot presents patient survival with expression of *ZG16*. The data from (**E**,**F**) was obtained from Gene Expression Profiling Interactive Analysis GEPIA web server [63].

**Figure 3 cancers-13-00049-f003:**
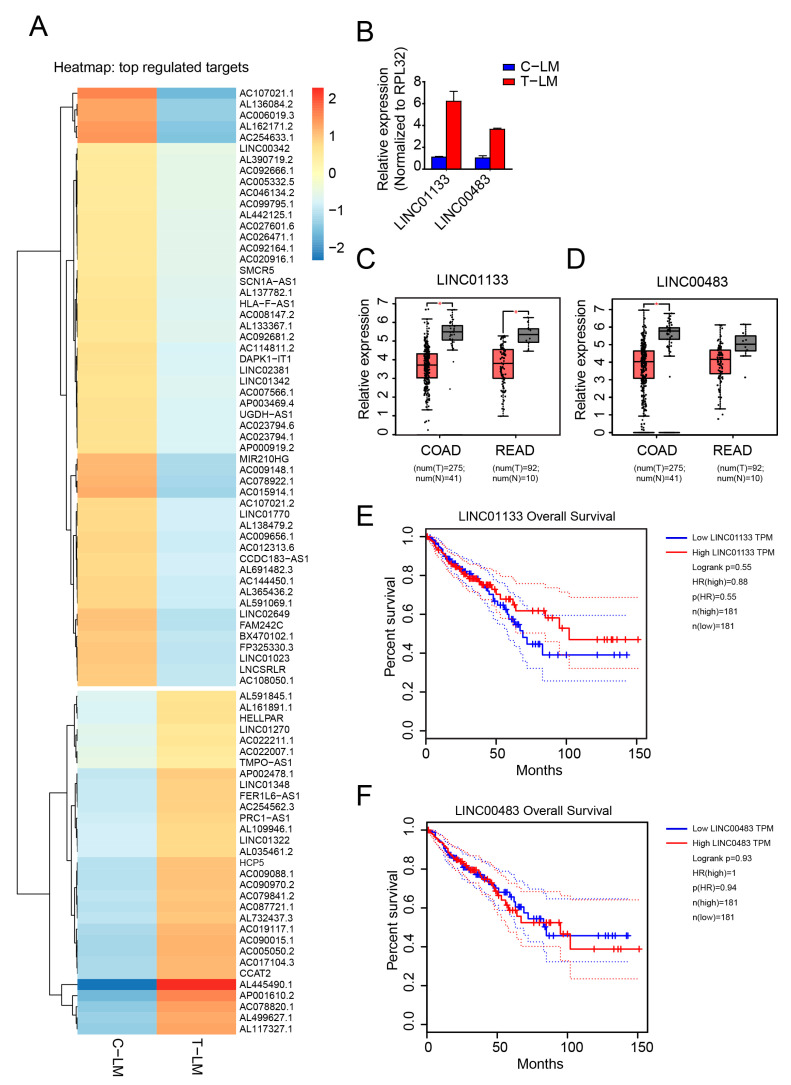
Identification of lncRNAs regulated during LDM topotecan chemotherapy. (**A**) Heatmap of differentially expressed miRNAs in liver metastases from treated mice (T-LM) versus liver metastases from control mice (C-LM). (**B**) Validation of *LINC01133* and *LINC00483* as downregulated lncRNA from treated liver metastases compared to control liver metastases by qPCR. (**C**,**D**) Boxplot showing the significant downregulation of *LINC01133* and *LINC00483* in colon adenocarcinoma (COAD) and rectum adenocarcinoma (READ) obtained from TCGA data, respectively. (**E**,**F**) Survival plot presents patient survival with expression of *LINC01133* and *LINC00483* in colon adenocarcinoma (COAD) and rectum adenocarcinoma (READ) obtained from TCGA data, respectively. The data from (**C**–**F**) was obtained from Gene Expression Profiling Interactive Analysis GEPIA web server [63].

**Figure 4 cancers-13-00049-f004:**
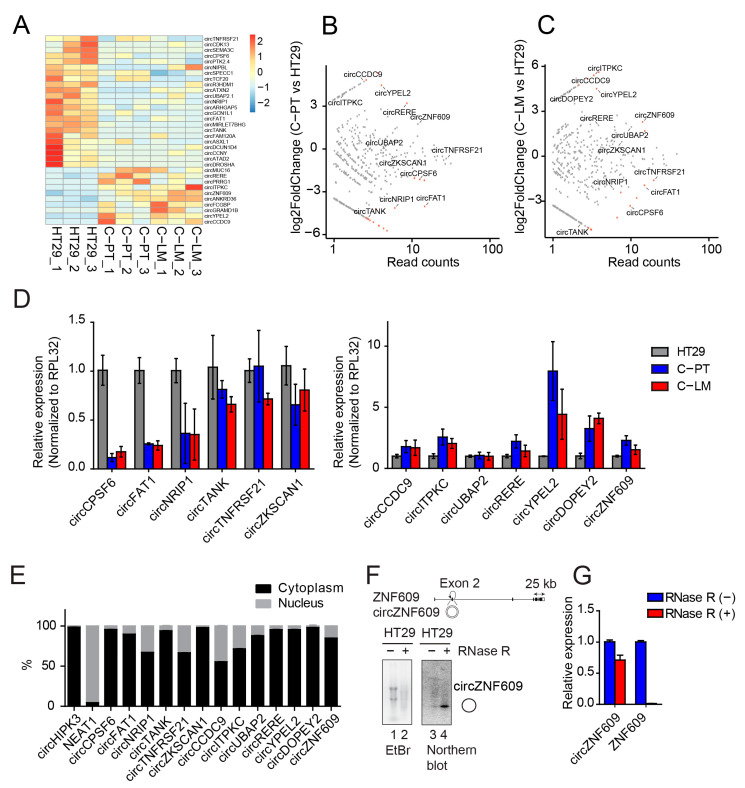
Identification of differentially expressed circRNAs during cancer progression and characterization of circZNF609. (**A**) Heatmap of differentially expressed circRNAs in primary tumors (C-PT), liver metastases (C-LM) versus HT29. The total number of reads supporting a particular head-to-tail junction was used as an absolute measure of circRNA abundance. (**B**,**C**) Scatter plot of circRNA expression in C-PT versus HT29 and C-LM versus HT29, respectively. (**D**) Validation of differentially expressed circRNAs in C-PT and C-LM from different mice compared to HT29. (**E**) Quantification of the nuclear/cytoplasmic localization of validated circRNAs. (**F**) Scheme of *ZNF609* locus with circZNF609 generated from its exon 2. The circularized exon 2 of circZNF609 is located between very long introns as observed for several circRNAs. Total RNA from HT29 treated with RNase R (lane 2) or mock treated (lane 1) were loaded on agarose gel. To enhance the signal of circZNF609, more RNA material was loaded in treated samples (lane 2, 4) compared to the untreated sample (lane 1, 3). Northern blot analysis of circZNF609 after RNase R treatment (lane 4) showed significant levels of circZNF609. (**G**) Relative expression of circZNF609 and ZNF609 using RNA isolated from HT29 cells treated with and without RNase R.

**Figure 5 cancers-13-00049-f005:**
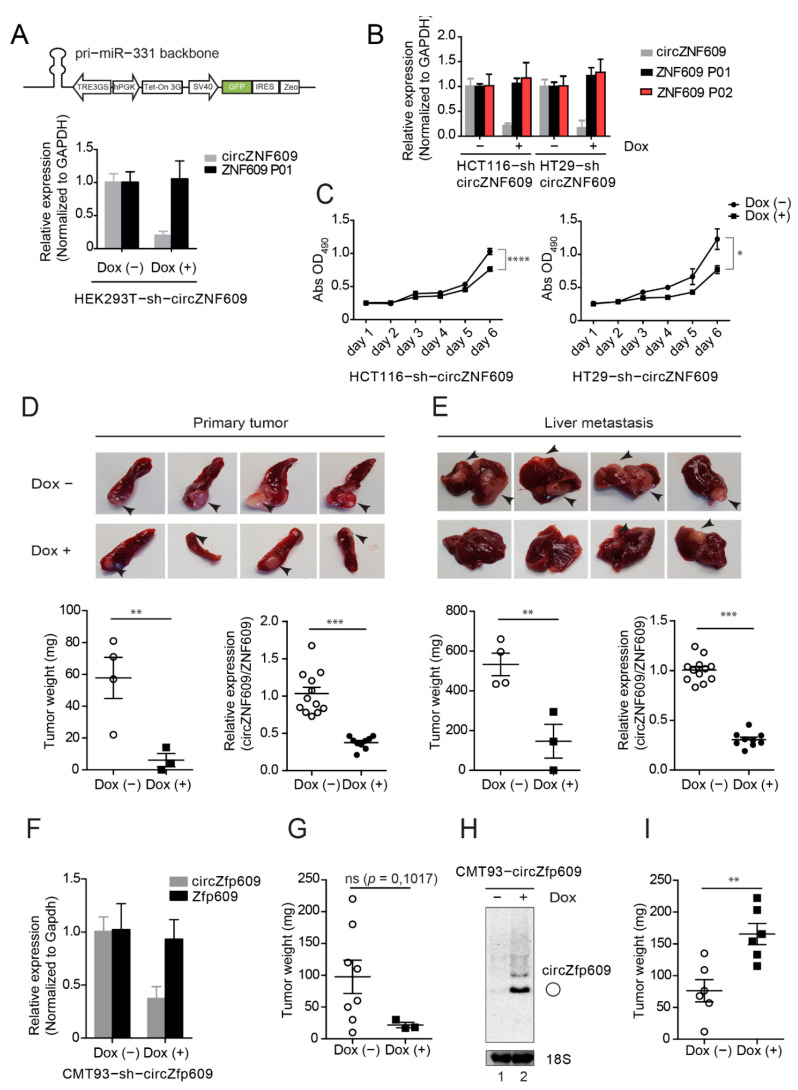
circZNF609 promotes colorectal cancer development. (**A**) shRNA design for knocking down of circZNF609 based on pri-miR-331 cassette driven by Poll-II promoter. shRNA targeting circZNF609 was designed by replacing miR-331 sequence. Using this construct, circZNF609 is efficiently knocked down in HEK293T-sh-circZNF609 upon doxycycline induction. (**B**) Relative expression of circZNF609 and ZNF609 (using two different primer pairs P01 and P02) in HT29-sh-circZNF609 and HCT116-sh-circZNF609 upon doxycycline induction validated by qPCR. (**C**) Knockdown of circZNF609 decreases cell proliferation in both HT29-sh-circZNF609 and HCT116-sh-circZNF609 cell lines. (**D**,**E**) HT29-sh-circZNF609 cell lines were injected into spleen of the SCID mice. Spleens containing primary tumors (PT) and livers with metastases (LM) were imaged for macro analysis. The according tumors were resected and measured by tumor weight (*n* = 8). Relative expression of circZNF609 of RNA isolated from PT and LM by qPCR, which was normalized to ZNF609. (**F**) Relative expression of circZfp609 in CMT93-sh-circZfp609 upon doxycycline induction validated by qPCR. (**G**) In vivo data of tumor resected from the mice subcutaneously injected with CMT93-sh-circZfp609 upon doxycycline induction. The tumor development was calculated by measuring the tumor weight (*n* = 11). (**H**) CMT93-circZfp609 cell was induced by doxycycline and the cells were harvested for northern blot analysis. (**I**) In vivo data from the mice subcutaneously injected of CMT93-circZfp609 cells upon doxycycline induction (*n* = 12). The tumor development was calculated by measuring the tumor weight. Dox: Doxycycline. ns: not significant, *p* > 0.05; * *p* ≤ 0.05; ** *p* ≤ 0.01; *** *p* ≤ 0.001.

## Data Availability

RNAseq data is available from Gene Expression Omnibus (GEO) under the accession number GSE163076. Uncropped western blots and northern blots can be found in Appendix A.

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
