# Peer review of "Gene Expression Signatures of a Preclinical Mouse Model during Colorectal Cancer Progression under Low-Dose Metronomic Chemotherapy"

_cancers, 2020, doi:10.3390/cancers13010049_

Round 1

Reviewer 1 Report

In this experimental study, the authors investigate possible gene signatures of colorectal cancer progression under LDM therapy to find new potential targets for therapeutic interventions.

Overall, the study seems to be well conducted and reports interesting and relevant results. The data is well presented. I have no major concerns.

Minor concerns:

  • While I am aware of Dr. Hackl’s 2013 Gut paper, topotecan has not been introduced into clinical practice for colorectal cancer since then. Of course this may be due to multiple reasons and does not necessarily reflect limited efficacy of topotecan in real-life CRC. However, the authors should explain their choice of topotecan over the multitude of other chemotherapeutics clinically used in CRC.
  • Many authors regard splenic injection models as a “CTC homing model” rather than a real metastasis model as the injected tumor cells probably directy disseminate into the liver (as opposed to “real” metastasis models, requiring tumor cells to invade the blood stream prior to dissemination to distant organs). I believe the choice of model is still suitable for the experimental question; however, the authors should explain their choice of mouse in the introduction and discuss its limitations.
  • Different tenses are used, even in one sentences (e.g.: lines 499-501 „Primary tumor (…) and metastases in the liver resections were performed when the tumor size is big enough or when the bioluminescence measurement is significant.“) Please check the text for inconsistent tense use.
  • Lines 105 – 119: Reporting of results should take place in the results section.
  • Numbers up to 12 are normally spelled out (line 125: three to seven days (instead of 3-7 days), line 127: six to eight weeks)
  • Simple Summary: Something went wrong in the first sentence - please rephrase.
  • Typos:
    1. Line 62: suggested as (instead of „suggestedas“)
    2. Line 144: it’s (instead of „it“)
    3. Line 179: was obtained (instead of „was obtain“)
    4. Line 255: regulated in a similar way (instead of „in as similar way“)
    5. Line 333: conserved in mice (instead of „mouse“)
    6. Line 413: upregulated (instead of „up regulated“)
    7. Line 632: eppendorf tube (instead of „eppendoft tube“)
    8. Line 651: a single-cell based analysis (instead of „analyses“)

Author Response

Reviewer 1:

In this experimental study, the authors investigate possible gene signatures of colorectal cancer progression under LDM therapy to find new potential targets for therapeutic interventions. Overall, the study seems to be well conducted and reports interesting and relevant results. The data is well presented. I have no major concerns.

Minor concerns:

While I am aware of Dr. Hackl’s 2013 Gut paper, topotecan has not been introduced into clinical practice for colorectal cancer since then. Of course this may be due to multiple reasons and does not necessarily reflect limited efficacy of topotecan in real-life CRC. However, the authors should explain their choice of topotecan over the multitude of other chemotherapeutics clinically used in CRC.

Topotecan is an inhibitor of the DNA-topoisomerase 1. The same mechanism of action is used by the DNA-topoisomerase inhibitor irinotecan, which is one of the standard chemotherapeutic drugs given to colorectal cancer patients. The reason why we used topotecan instead of irinotecan is, that in contrast to irinotecan, topotecan can be given orally, which is a great benefit for a daily, low-dose application both in preclinical trials as well as in patient treatment. This information has now been added into the manuscript (page 2, line 55-59).

While topotecan is a standard drug for treating ovarian cancer and small cell lung cancer, it is correct, that it has so far not been introduced into clinical practice in colorectal cancer. The research group of Dr Hackl has since publication of their 2013 GUT paper worked on the clinical translation of oral topotecan treatment for CRC, but sadly, since the drug is “old”, cheap, and no longer protected by patents, no sufficient financial support has been gained in spite of several good reviews by non-industrial funding agencies.

Many authors regard splenic injection models as a “CTC homing model” rather than a real metastasis model as the injected tumor cells probably directy disseminate into the liver (as opposed to “real” metastasis models, requiring tumor cells to invade the blood stream prior to dissemination to distant organs). I believe the choice of model is still suitable for the experimental question; however, the authors should explain their choice of mouse in the introduction and discuss its limitations.

We absolutely agree with the reviewer, that the splenic injection model in in large parts a CTC homing model rather than a metastasis model. Actually, our group has performed extensive analyses injecting luciferase-tagged cancer cells into the spleen and imaging after several seconds/minutes/hours/days, partly with splenectomy immediately after cancer cell injection, partly without. We could show that immediately after injection, fluorescence signalling can be seen in the liver. However, a bigger hepatic tumor load is achieved in mice not undergoing splenectomy but developing a large splenic “primary” tumor several weeks after splenic injection. Thus, we and others use this model as simple and early metastasis model but discuss its limitations accordingly. (Hackl et al., Gut 2013; Soares et al., J Vis Exp 2014; Morin et al., Oncotarget 2017; Dafflon et al, Intestinal Sem Cells 2020;) Of course, we also apply orthotopic intracecal injection studies which is also the next step planned with our results obtained in the present manuscript. We have added this information and discussion in the introduction (page 3, line 113-114) and discussion (page 14, lines 454-455). Even before revision, it was mentioned in the conclusion (lines 780-785).

Different tenses are used, even in one sentences (e.g.: lines 499-501 „Primary tumor (…) and metastases in the liver resections were performed when the tumor size is big enough or when the bioluminescence measurement is significant.”) Please check the text for inconsistent tense use.

We thank the reviewer for this remark. The whole manuscript has now been carefully checked and inconsistent tense uses have been corrected.

Lines 105-119: Reporting of results should take place in the results section

We thank the reviewer for this suggestion. However, we thought that a very short summary of these findings helps the readability of the introduction. We now have added an explanatory “in this study” before this paragraph (now page 3, lines 111-127) and would prefer to keep this paragraph. However, if the scientific editors would also prefer us to delete or shorten this paragraph in the introduction, we will do so.

Numbers up to 12 are normally spelled out (line 125: three to seven days (instead 3-7 days), line 127: six to eight weeks).

The whole manuscript has now been carefully checked and number up to 12 are now spelled out.

Simple Summary: Something went wrong in the first sentence – please rephrase.

This sentence has now been corrected (page 1, line 16-17)

Typos:

  1. Line 62: suggested as (instead of „suggestedas")

This typo has been corrected (now line 67)

  1. Line 144: it’s (instead of „it“)

This typo has been corrected (now line 156)

  1. Line 179: was obtained (instead of „was obtain“)

This typo has been corrected there (now line 191) and also in line 202.

  1. Line 255: regulated in a similar way (instead of „in as similar way“)

This typo has been corrected (now line 267)

  1. Line 333: conserved in mice (instead of „mouse“)

This typo has been corrected (now line 350)

  1. Line 413: upregulated (instead of „up regulated“)

This typo has been corrected (now line 438)

  1. Line 632 eppendorf tube (instead of „eppendoft tube“)

This typo has been corrected (now line 675)

  1. Line 651: a single-cell based analysis (instead of „analyses“)

This typo has been corrected (now line 694)

Reviewer 2 Report

The manuscript by Hung Ho-Xuan et al. entitled: “Gene expression signatures of a preclinical mouse model during colorectal cancer progression under low-dose metronomic chemotherapy” investigate the transcriptomic changes that occur during colorectal cancer progression and how expression signatures are affected by low dose metronomic (LDM) therapy. This was done in a comprehensive manner using RNA sequencing, which allowed for mRNAs as well as non-coding RNAs such as microRNAs, long non-coding RNAs and circular RNAs to be profiled. In addition, the authors find that circZNF609 functions as oncogene since overexpression studies lead to an increased tumor growth while specific knock down resulted in smaller tumors in the mice. However, the authors do not investigate the mechanisms of action of circZNF609. Overall, I find that the topic of the article is highly relevant and addressed by appropriate experiments. However, the manuscript could be significantly improved by inclusion of primary patient samples, but I acknowledge that this may not be feasible to do in a revision if no ethical permission has been obtained for this purpose. I find that the data are generally well presented and that they support the conclusions of the paper, which is well written. I have the following suggestions for the authors to consider before the paper is published:

  1. Lines 102-104: The following sentence needs to be backed up by a reference: “Strikingly, knockdown screening of a large panel of circRNAs revealed that more than 10% are important for cell proliferation while their parental, linear transcripts are not”.
  2. Lines 109-110: “we found that a group of interferon-induced genes are most upregulated in expression”. Please consider rephrasing to something like: “we found that the expression of a group of interferon-induced genes were most upregulated”.
  3. Figure 1A: at what time point was the RNA collected? Both at day1 week1 and after 6-8 weeks? The data presented in the remaining panels of figure 1- what time points were analyzed? I guess these data are from week 6-8? I think it may help the reader to be more clear about this in the figure legend and to change the title of the legend so that it better reflects the nature of the data/findings. Also, I guess the title should be in bold as for the other figure legends? Please also re-consider the title of figure legend 4.
  4. In line 335: “500 000” should be “500,000”.
  5. Please ensure correct usage of circZfp609/circZNF609 throughout the manuscript.
  6. Ideally, the authors should perform in situ analysis of circZNF609 in colorectal cancer specimens to prove that it is present in the cancer cells of patient tumors. It has recently been shown that this is not the case in colorectal cancer for another highly studied circRNA in cancer (CDR1as/ciRS-7) (Nat Commun. 2020 Sep 11;11(1):4551. doi: 10.1038/s41467-020-18355-2) despite numerous studies have indicated an oncogenic role for this circRNA through in vitro and in vivo studies. If it is not possible at this time to include patient samples in the study, the authors should discuss these important findings and mention that future studies of circZNF609 should assess its spatial expression patterns within primary tumors.
  7. Please put gene names in italics.
  8. Please carefully check the entire manuscript to correct typos and minor grammatical errors.
  9. Please check all references carefully. Several seem to miss page numbers.

Author Response

Reviewer 2

The manuscript by Hung Ho-Xuan et al. entitled: “Gene expression signatures of a preclinical mouse model during colorectal cancer progression under low-dose metronomic chemotherapy” investigate the transcriptomic changes that occur during colorectal cancer progression and how expression signatures are affected by low dose metronomic (LDM) therapy. This was done in a comprehensive manner using RNA sequencing, which allowed for mRNAs as well as non-coding RNAs such as microRNAs, long non-coding RNAs and circular RNAs to be profiled. In addition, the authors find that circZNF609 functions as oncogene since overexpression studies lead to an increased tumor growth while specific knock down resulted in smaller tumors in the mice. However, the authors do not investigate the mechanisms of action of circZNF609. Overall, I find that the topic of the article is highly relevant and addressed by appropriate experiments. However, the manuscript could be significantly improved by inclusion of primary patient samples, but I acknowledge that this may not be feasible to do in a revision if no ethical permission has been obtained for this purpose. I find that the data are generally well presented and that they support the conclusions of the paper, which is well written.

We thank the reviewer very much for these comments. It is correct, that we do not investigate the mechanisms of action of circZNF609 in this manuscript. As we mentioned in the discussion and now have even stressed this point circZNF609 is not translated. We also screened for microRNA binding sites, but up until now cannot yet say which function miRNA has to circZNF609 (see lines 505-511),.

 I have the following suggestions for the authors to consider before the paper is published:

  1. Lines 102-104: The following sentence needs to be backed up by a reference: “Strikingly, knockdown screening of a large panel of circRNAs revealed that more than 10% are important for cell proliferation while their parental, linear transcripts are not”.

We thank reviewer 2 for pointing out this very important aspect. We added the reference corresponding to this sentence (line 109).

  1. Lines 109-110: “we found that a group of interferon-induced genes are most upregulated in expression”. Please consider rephrasing to something like “we found that the expression of a group of interferon-induced genes were most upregulated”

We thank the reviewer for this suggestion and have changed this sentence accordingly (lines 117-118)

  1. Figure 1A: at what time point was the RNA collected? Both at day1 week1 and after 6-8 weeks? The data presented in the remaining panels of figure 1 – what time points were analyzed? I guess these data are from week 6-8? I think it may help the reader to be more clear about this in the figure legend and to change the title of the legend so that it better reflects the nature of the data/findings. Also, I guess the title should be in bold as for the other figure legends? Please also re-consider the title of figure legend 4.

We thank the reviewer improving the readability of our manuscript and figures. Indeed, we collected the RNA from weeks 6-8 when we dissected the tumors. We have changed this in the legend of Figure 1A to make this point clear (lines 140-148). We also changed the titles of Figure 1 (line 140) and Figure 4 (line 289).

  1. In line 335: “500 000” should be “500,000”.

This typo has been corrected here (now line 352) as well as throughout the manuscript.

  1. Please ensure correct usage of circZfp609/circZNF609 throughout the manuscript

The correct spelling of circZfp609/circZNF609 has now been checked throughout the manuscript.

  1. Ideally, the authors should perform in situ analysis of circZNF609 in colorectal cancer specimens to prove that it is present in the cancer cells of patient tumors. It has recently been shown that this is not the case in colorectal cancer for another highly studied circRNA in cancer (CDR1as/ciRS-7)(Nat Commun. 2020 Sept 11;11(1):4551.doi:10.1038/s41467-020-18355-2) despite numerous studies have indicated an oncogenic role for this circRNA through in vitro and in vivo studies. If it is not possible at this time to include patient samples in the study, the authors should discuss these important findings and mention that future studies of circZNF609 should assess its spatial expression patterns with primary tumors.

We regret that in the current manuscript, we will not be able to include patient samples. We have modified the discussion accordingly (now lines 444-459). Even before revision, this was mentioned in the conclusion (now lines 689-698).

  1. Please put gene names in italics

Gene names have now been put in italics throughout the manuscript.

  1. Please carefully check the entire manuscript to correct typos and minor grammatical errors

The entire manuscript has carefully been checked, typos and grammatical errors have been corrected.

  1. Please check all references carefully. Several seem to miss page numbers.

All references have been checked and corrected if necessary.